# The oocyte zinc transporter *Slc39a10/Zip10* is a regulator of zinc sparks during fertilization in mice

**Atsuko Kageyama[1], Narumi Ogonuki[2], Takuya Wakai[3], Takafumi Namiki[1†], Yui Kawata[1], Manabu Ozawa[4], Yasuhiro Yamada[5], Toshiyuki Fukada[6], Atsuo Ogura[2], Rafael Fissore[7], Naomi Kashiwazaki[1,8], Junya Ito[1,8,9]***

[1]Laboratory of Animal Reproduction, School of Veterinary Medicine, Azabu University, Sagamihara, Japan; [2]Bioresource Engineering Division, Bioresource Research Center, RIKEN, Tsukuba, Japan; [3]Department of Animal Science, Graduate School of Environmental and Life Science, Okayama University, Okayama, Japan; [4]Laboratory of Reproductive Systems Biology, Center for Experimental Medicine and Systems Biology, The Institute of Medical Science, The University of Tokyo, Minato-ku, Japan; [5]Department of Molecular Pathology, Graduate School of Medicine, The University of Tokyo, Bunkyo-ku, Japan; [6]Laboratory of Molecular and Cellular Physiology, Faculty of Pharmaceutical Sciences, Tokushima Bunri University, Tokushima, Japan; [7]Department of Veterinary and Animal Sciences, University of Massachusetts Amherst, Amherst, United States; [8]Graduate School of Veterinary Science, Azabu University, Sagamihara, Japan; [9]Center for Human and Animal Symbiosis Science, Azabu University, Sagamihara, Japan

**\*For correspondence:**
itoj@azabu-u.ac.jp

**Present address:** †Department of Life Science Frontiers, Center for iPS Cell Research and Application (CiRA), Kyoto University, Kyoto, Japan

**Competing interest:** The authors declare that no competing interests exist.

## eLife Assessment

This study presents significant and novel insights into the roles of zinc in mammalian meiosis/fertilization events. These findings are **useful** to our understanding of these processes. The evidence presented is **solid**, with experiments being well-designed, carefully described, and interpreted with appropriate rigor. The authors acknowledge the lack of mechanistic insight which represents the main limitation of the study.

**Abstract** In all vertebrates studied to date, a rise(s) in intracellular calcium is indispensable for successful fertilization and further embryonic development. Recent studies demonstrated that zinc is ejected to the extracellular milieu, the 'zinc spark', and follows the first few calcium rises of fertilization. However, the role of the zinc sparks in fertilization and development, as well as the supporting influx mechanism(s), remains unknown. In this study, we focused on zinc transporters *Slc39a10/Zip10* which were expressed in mouse oocytes through follicular development and investigated the oocyte-specific deficient mice for *Slc39a10/Zip10* (*Slc39a10* cKO: *Slc39a10flox/flox Gdf9Cre/+*). *Slc39a10* mRNA or SLC39A10/ZIP10 protein was expressed throughout folliculogenesis in the oocyte or plasma membrane, respectively. The number of ovulated oocytes was examined in *Slc39a10* cKO mice, and no change from the number of oocytes was observed. *Slc39a10* cKO oocytes decreased zinc level in the oocytes but did not affect maturation and metaphase II spindle formation. Fertilization-induced calcium oscillations were present in *Slc39a10* cKO oocytes, but zinc sparks were not observed. Despite other events of egg activation proceeding normally in *Slc39a10* cKO oocytes, embryo development into 4 cells and beyond was compromised. We show here for the first time that the zinc transporter ZIP10 contributes

to zinc homeostasis in oocytes and embryos, highlighting the role of labile zinc ions in early development.

## Introduction

In mammalian fertilization, the sperm factor is released into the ooplasm to activate the oocyte (*Yoon and Fissore, 2007*). It is now believed that phospholipase C zeta (PLC $\zeta$ ) (*Yoon and Fissore, 2007*; *Saunders et al., 2002*) is the sperm factor and it induces in the ooplasm repetitive increases in the intracellular $Ca^{2+}$, termed 'calcium oscillations'. These oscillations play an essential role in triggering the oocyte activation events, such as cortical granule exocytosis, the block of polyspermy, meiotic resumption, and exit from metaphase II (MII) arrest (*Xu et al., 1994*; *Jones, 2005*; *Ducibella et al., 2006*; *Ito et al., 2011*; *Sugita et al., 2024*). A calcium increase is highly conserved in many species as the trigger of egg activation and is vital for successful fertilization.

In recent years, zinc ions, as well as calcium ions, are thought to play important roles during fertilization (*Suzuki et al., 2010*; *Allouche-Fitoussi and Breitbart, 2020*; *Kageyama et al., 2022a*). Zinc ion is an essential trace element and the second most abundant transition ion in the human body after iron (*Lubna and Ahmad, 2023*). Zinc homeostasis is essential for optimal metabolic function in the reproductive process in mammals (*Pascua et al., 2020*); therefore, zinc deficiency causes abnormalities such as fetal teratogenicity, long gestation periods, problematic labor, low birth weight, and weak offspring (*Favier, 1992*; *Bedwal and Bahuguna, 1994*). Zinc also has an important role in the male and female germ cells (*Allouche-Fitoussi and Breitbart, 2020*; *Kageyama et al., 2022b*). Especially in the mouse female germ cells, zinc-deficient condition caused profound defects in oocyte maturation, cumulus cell expansion, and ovulation cycle (*Tian and Diaz, 2012*). Several studies have been conducted in mice to understand how zinc regulates oocyte maturation. Zinc was reported to be a critical regulator of meiosis throughout oocyte maturation, including maintaining release from the first and second meiotic arrest (*Kong et al., 2012*). In addition, zinc levels of oocytes were found to increase by 50% as the oocyte progressed through meiosis (*Kim et al., 2010*; *Bernhardt et al., 2011*).

In 2011, Kim et al. reported the importance of zinc during mammalian fertilization (*Kim et al., 2011*). Zinc is predominantly stored in vesicles that are symmetrically arranged along the oocyte cortex at the germinal vesicle (GV) stage, and zinc-stored vesicles (cortical granules) are located away from the spindle and form a hemispherical pattern at the MII stage (*Kim et al., 2011*; *Que et al., 2015*; *Kong et al., 2014*; *Que et al., 2019*; *Jo et al., 2019*; *Que et al., 2017*). After the penetration of sperm, it has been induced a 'zinc spark' that releases billions of labile zinc ions from the cortical granules, and the total zinc content of the oocytes decreases by 10–20% at the end of fertilization in mice (*Kim et al., 2011*; *Lee et al., 2020*). In addition, these events immediately follow a series of calcium oscillations (*Kim et al., 2011*). Zinc spark has also been observed in the cattle, nonhuman primate, human, and *Xenopus laevis* oocytes (*Kim et al., 2011*; *Que et al., 2019*; *Duncan et al., 2016*; *Seeler et al., 2021*), suggesting that this phenomenon is a highly conserved event in vertebrates. More surprisingly, full-term development of mouse embryos has been reported by chelation of $Zn^{2+}$ ions without $Ca^{2+}$ release, suggesting the depletion of zinc ions in oocytes may be sufficient for oocyte activation (*Suzuki et al., 2010*). Zinc has also been reported to impact the sperm during fertilization in mice. Zinc accumulation in the zona pellucida increases fibril binding along the glycoprotein matrix and decreases the number of sperm that can reach the fertilized oocytes (*Que et al., 2017*; *Aonuma et al., 1981*; *Tokuhiro and Dean, 2018*). After the zinc sparks, the released zinc affects the forward motility of sperm to prevent their passage through the zona matrix (*Tokuhiro and Dean, 2018*). Zinc spark is associated with the release of ovastacin (*Astl*, the official gene name), which is required for the zona reaction and functions as a polyspermy block mechanism that is initiated only a few minutes after fertilization (*Tokuhiro and Dean, 2018*).

The transport of zinc ions into and out of cells is regulated by zinc transporters. In mammals, 14 zinc ion importers, called SLC39A/ZIP (ZIP1-ZIP14), have been identified (*Fukada and Kambe, 2011*; *Takagishi et al., 2017*; *Kambe et al., 2015*). Previous study showed that mammalian oocytes regulate zinc uptake through two maternally derived and cortically distributed zinc transporters, ZIP6 and ZIP10 (*Kong et al., 2014*). Further, they reported that targeted disruption using *Slc39a6-* and *Slc39a10*-specific morpholino injection or antibody incubation induced alteration of the intracellular labile zinc content, spontaneous resumption of meiosis from the PI arrest, and premature arrest at

a telophase I-like state (*Kong et al., 2014*). It is clear from these reports that ZIP6 and ZIP10 are involved in zinc transport in oocytes, but the function is not elucidated.

In this study, we generated oocyte-specific *Slc39a6* and *Slc39a10* conditional knockout mice and examined the function of ZIP10 in the oocytes and the importance of zinc homeostasis during fertilization and embryonic development. This study provides clues that elucidate its role in fertilization and embryonic development, which is still largely unknown. In addition, this is the first report confirming the function of the zinc transporter in oocytes, which will contribute to future research.

## Results

### Zinc transporters, ZIP6 and ZIP10, are expressed in mouse oocytes through follicular development

First, using in situ hybridization and immunofluorescent (IF) staining, we examined the expression of ZIP10 during follicular development. As shown in *Figure 1A*, *Slc39a10* mRNA was expressed from the primordial oocyte (arrow). It continued to be expressed in oocytes of primary, secondary, and antral follicles. ZIP10 protein was also expressed in the plasma membrane of primordial oocytes (*Figure 1B*; arrow). We also confirmed the expression of ZIP10 protein at the plasma membrane of oocytes of primary, secondary, and antral follicles. Although ZIP6 was also expressed in oocytes throughout folliculogenesis (*Figure 1C*; arrow), it displayed nuclear localization in oocytes and granulosa cells of primary, secondary, and antral follicles. *Figure 1D* showed zona pellucida and granulosa cells through the follicular development.

### Phenotype of oocyte-specific *Slc39a6* cKO and *Slc39a10* cKO female mice

To elucidate the roles of *Slc39a6* and *Slc39a10* in the mouse oocytes, oocyte-specific *Slc39a6* (*Slc39a6* cKO) and *Slc39a10* (*Slc39a10* cKO) knockout mice were generated (*Figure 2—figure supplement 1A–C*). We examined whether ZIP6 and ZIP10 protein expression was absent in *Slc39a6* cKO and *Slc39a10* cKO oocytes (*Figure 2—figure supplement 1D*). *Slc39a6*$^{f/f}$ and *Slc39a10*$^{f/f}$ mice were used as controls, respectively. We examined the number of ovulated oocytes or defects in oocyte maturation. After superovulation, the number of ovulated oocytes collected from the oviduct in *Slc39a10* cKO mice was 22.1±2.5 oocytes/mouse (*Figure 2A*), which was equivalent to that in *Slc39a10*$^{f/f}$ mice (18.7±2.9 oocytes/mouse, p>0.05). The numbers of ovulated oocytes in *Slc39a6* cKO and *Slc39a6*$^{f/f}$ mice were also equivalent (28.7±3.6 and 32.8±3.3 oocytes/mouse, respectively, p>0.05) (*Figure 2—figure supplement 2A*). As for oocyte maturation, the rate of oocytes with a first polar body at 10 hr, 12 hr, and 14 hr in the *Slc39a10* cKO group was not different than that of *Slc39a10*$^{f/f}$ group (*Slc39a10* cKO: 60.0%, 67.0%, and 79.0%, *Slc39a10*$^{f/f}$: 65.8%, 70.3%, and 80.2%, respectively, p>0.05, *Figure 2B*). In addition, *Slc39a10* cKO oocytes display intact metaphase spindles just as in the *Slc39a10*$^{f/f}$ groups (*Figure 2C*).

Next, we examined the levels of labile zinc in GV, MII, and two pronuclei (2PN) zygotes by comparing the fluorescence intensity following loading with the dye, FluoZin-3AM (*Figure 2D*). The FluoZin-3AM fluorescence intensity in GV and MII oocytes of the *Slc39a10* cKO group was lower than the *Slc39a10*$^{f/f}$ group (p<0.05). After fertilization, the fluorescence intensity in *Slc39a10*$^{f/f}$ zygotes decreased dramatically and also decreased for the *Slc39a10* cKO group, although it remained higher than for *Slc39a10*$^{f/f}$ oocytes. When compared within group, the fluorescence intensity in *Slc39a10*$^{f/f}$ between GV and MII oocytes was significantly different (GV vs. MII and MII vs. 2PN; p<0.05, respectively), but the stages of *Slc39a10* cKO oocytes were not different despite clear trends (p>0.05). The levels of zinc fluorescence intensity in the *Slc39a6* cKO group were not different from the controls *Slc39a6*$^{f/f}$, which decreased markedly at fertilization (*Figure 2—figure supplement 2B*; p>0.05).

### The zinc sparks of *Slc39a10* cKO oocytes were suppressed after fertilization or artificial activation

To determine if the absence of ZIP6 and ZIP10 influenced the detection of $Zn^{2+}$ sparks associated with fertilization, we monitored extracellular zinc and intracellular calcium during fertilization of *Slc39a6* cKO and *Slc39a10* cKO oocytes (*Figure 3A*, *Figure 3—figure supplement 1*). *Slc39a10*$^{f/f}$ oocytes displayed the expected calcium oscillations following fertilization, and a zinc spark followed

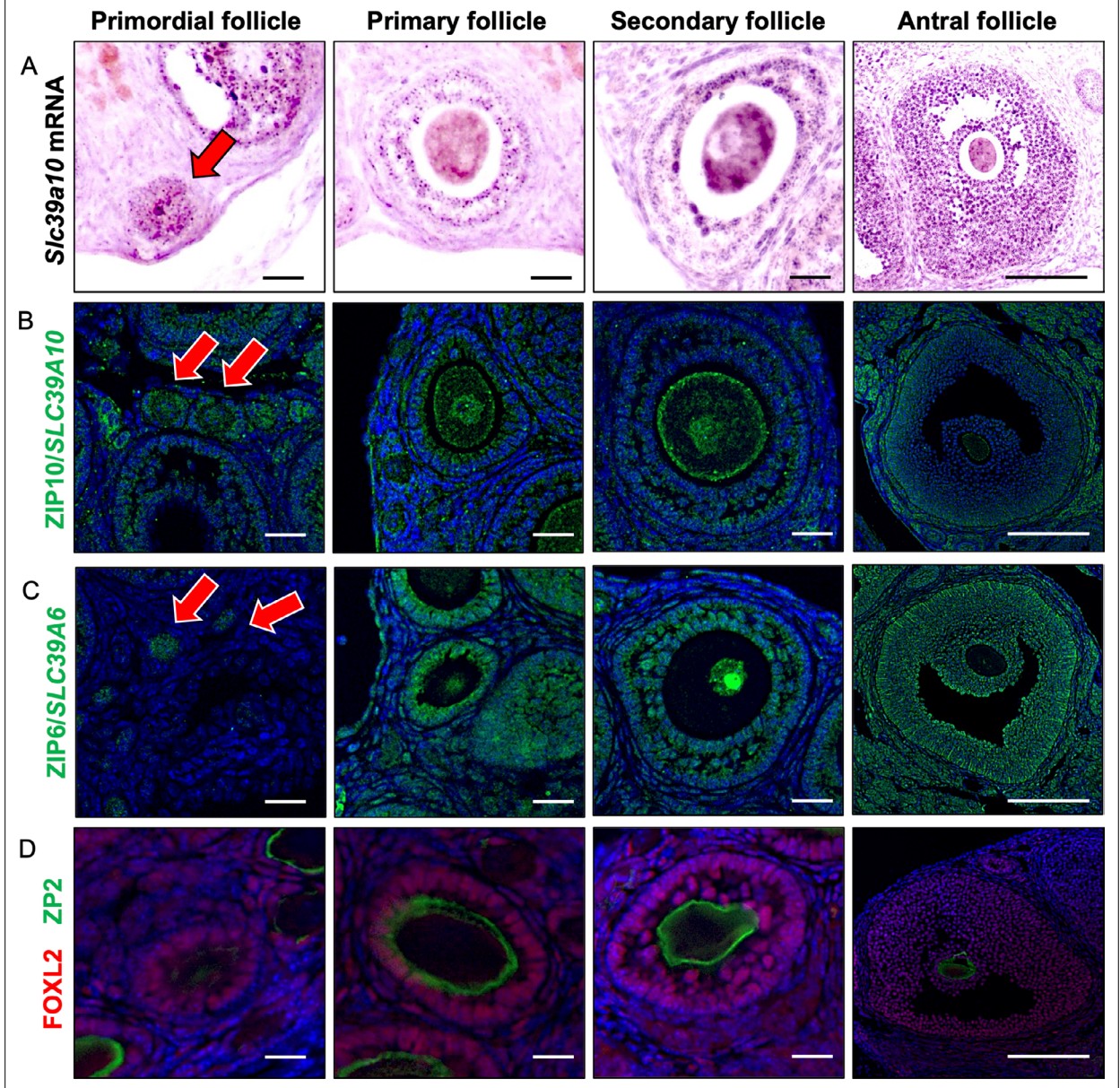

**Figure 1.** Expression of ZIP6 and ZIP10 in mouse ovary. (**A**) In situ hybridization in the mouse ovary showed ZIP10 expression in oocyte and granulosa cell from primordial, primary, secondary, and antral follicle. Arrow indicates primordial follicular oocyte. (**B**) Immunofluorescent staining for ZIP10 (green) in the mouse ovary showed ZIP10 expression in oocyte membrane. Arrow indicates primordial follicular oocyte. (**C**) Immunofluorescent staining for ZIP6 (green) in the mouse ovary showed ZIP6 expression in oocyte nucleus and granulosa cells. Arrow indicates primordial follicular oocyte. (**D**) Immunofluorescent staining showed ZP2 (green; zona pellucida) and FOXL2 (red; granulosa cells) in the mouse ovary. It was observed that ZP2 was not present in the primordial follicle; however, it was present in the primary, secondary, and antral follicles. Furthermore, FOXL2 was observed at granulosa cells of all stage follicles. Scale bar: 20 μm (primordial, primary, and secondary follicle); 150 μm (antral follicle) (**A–D**).

the first calcium rise (*Figure 3A* upper side, *Videos 1 and 2*). In contrast, *Slc39a10* cKO oocytes did not release zinc ions immediately after the first calcium spike, despite mounting normal calcium oscillations (*Figure 3A* lower side, *Videos 3 and 4*). In *Slc39a6* cKO oocytes, a zinc spark occurred immediately after the first intracellular calcium rise at fertilization, just as in control *Slc39a6*^{f/f} oocytes (*Figure 3—figure supplement 1*, *Figure 3—videos 1–4*). The extracellular zinc sparks were examined following artificial oocyte activation of mouse oocytes with ionomycin. The zinc sparks occurred immediately after the intracellular calcium rise in *Slc39a10*^{f/f} oocytes (*Figure 3B* upper side, *Videos 5 and 6*); however, *Slc39a10* cKO oocytes did not release zinc ions after the calcium spike (*Figure 3B* lower side, *Videos 7 and 8*).

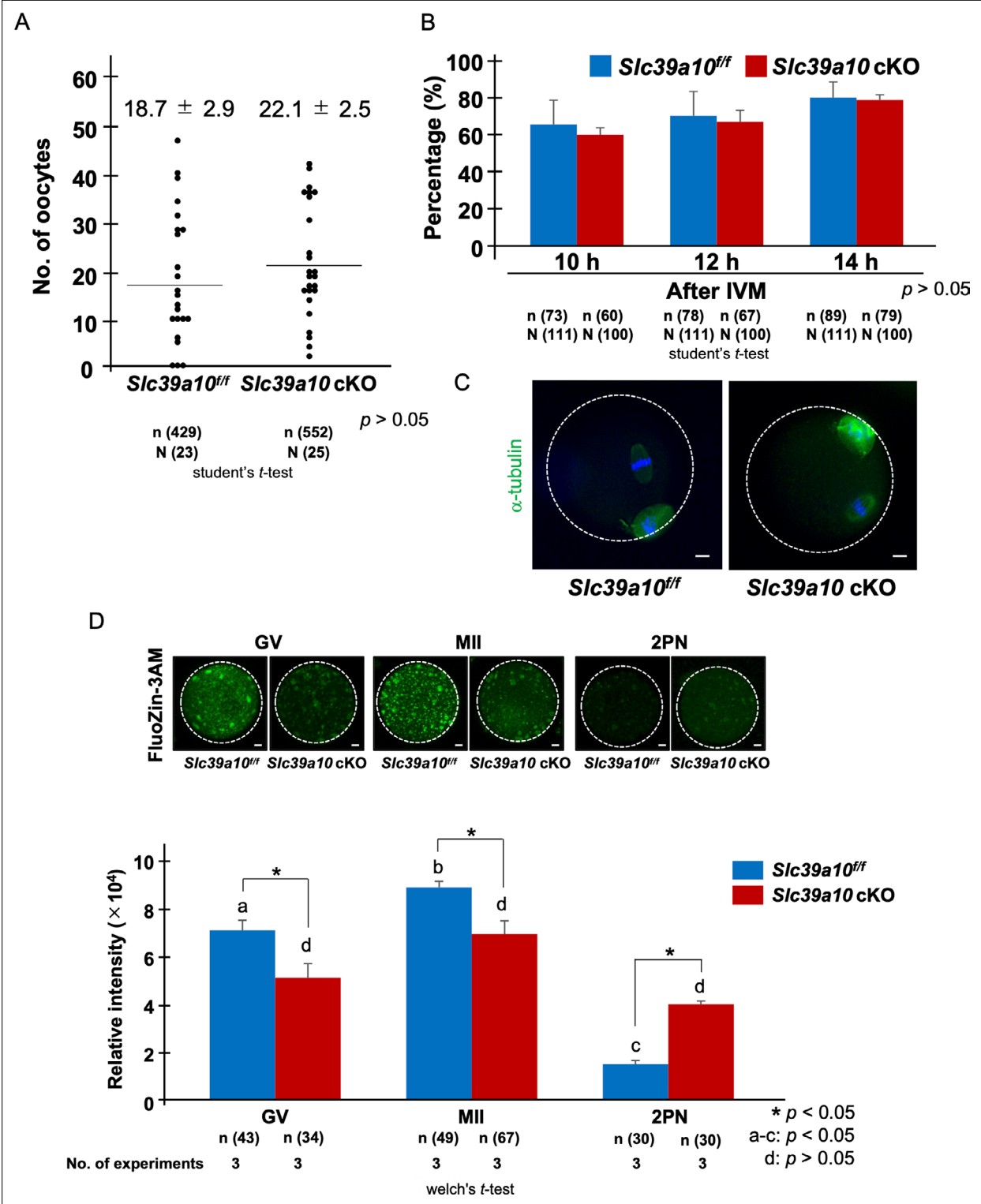

**Figure 2.** Number of collected oocytes and dynamics of labile zinc ion in *Slc39a10* cKO mice. (**A**) The results of average number of oocytes in each group. Data represents the average ± SEM. These experiments were repeated at least thrice. Statistical differences were calculated according to Student's *t*-test (p>0.05; no significant difference). (**B**) The percentage of extrusion of first polar body at 10, 12, and 14 hr after in vitro maturation (IVM). These experiments were repeated at least thrice. Statistical differences were calculated according to Student's *t*-test (p>0.05; no significant difference). (**C**) The morphology of spindle and chromosome organization in *Slc39a10*^f/f^ and *Slc39a10* cKO metaphase II (MII) oocytes at 14 hr after IVM. Anti-α-tubulin antibody (green) was used to stain the spindles. Chromosomes were stained with DAPI (blue). The scale bar represents 10 μm. (**D**) Comparison

*Figure 2 continued on next page*

*Figure 2 continued*

with the fluorescence intensity of intracellular labile zinc ion in germinal vesicle (GV), MII, and two pronuclei (2PN). The upper images showed the fluorescence of intracellular labile zinc ion of oocyte or embryo treated with 2 μM FluoZin-3AM for 1 hr. Representative images are shown. The white dotted circles indicate the positions of oocytes and embryos. Scale bars denote 10 μm. The lower part showed the fluorescence intensity of labile zinc ions in oocytes or embryos. Data represent the average ± SE of the experiments. For each experiment, 10–20 oocytes/embryo were stained and used for the measurement in each stage of the experiment, and these experiments were repeated three times. Statistical differences were calculated according to the Welch's *t*-test. Different letters represent significant differences (p<0.05).

The online version of this article includes the following source data and figure supplement(s) for figure 2:

**Figure supplement 1.** Generation of oocyte-specific *Slc39a6* and *Slc39a10* conditional knockout mouse.

**Figure supplement 1—source data 1.** PDF file containing original gels and western blots for *Figure 2—figure supplement 1C and D*, indicating the relevant bands and treatments.

**Figure supplement 1—source data 2.** Original files for original gel and western blot analysis displayed in *Figure 2—figure supplement 1C and D*.

**Figure supplement 2.** Number of collected oocytes, dynamics of labile zinc ion, and percentage of fertilization in *Slc39a6* cKO mice.

### *Slc39a10* cKO mouse oocytes can be fertilized but were unlikely to develop to the blastocysts

*Slc39a6* cKO and *Slc39a10* cKO oocytes were then used for IVF, and we examined the rates of fertilization, polyspermy, and embryo development. Successful fertilization was confirmed at 6 hr after IVF by the presence of pronuclei (*Figure 4A and B*). *Slc39a10* cKO oocytes were fertilized at rates like those observed for control oocytes (*Figure 4A*). Consistent with this, the zona reaction triggered by fertilization and examined using a ZP2 antibody was similar in *Slc39a10* cKO and *Slc39a10*^f/f oocytes (*Figure 4D*). We also examined the localization of ovastacin, whose expression in the cortex and post-fertilization loss was comparable between the two groups (*Figure 4E*). As for the expression of JUNO, it had the same expression as between null and control oocytes (*Figure 4—figure supplement 1*) and the temporal dynamics of its disappearance from the cortex after fertilization was similar for both *Slc39a10*^f/f and *Slc39a10* cKO groups (*Figure 4F*).

We next examined the development to the blastocyst stage, and whereas *Slc39a10*^f/f zygotes developed at the expected rates – approximately 78.9% (2-cell), 90.3 (4- to 8-cell), and 75.0% (blastocyst) – a smaller fraction of *Slc39a10* cKO zygotes did: 37.2% (2-cell), 38.3% (4- to 8-cell), and only 32.7% reached blastocyst stage (p<0.05; *Figure 4B*). As shown in *Figure 4C*, the total cell numbers of blastocysts were also lower for *Slc39a10* cKO embryos (51.6±2.1 cells) than for those of *Slc39a10*^f/f ones (72.7±2.1 cells, p<0.05). *Slc39a10* cKO oocytes did not display any alterations in the rate of fertilization or development (*Figure 4—figure supplement 2A and B*; p>0.05).

## Discussion

The zinc sparks associated with fertilization were first discovered in the mouse (*Kim et al., 2011*). Similar observations followed in other mammalian species and amphibians (*Que et al., 2019*; *Duncan et al., 2016*; *Seeler et al., 2021*). However, the underlying mechanism(s) and biological role of zinc sparks are not elucidated. In this study, we have examined the role of the zinc transporters ZIP6 and ZIP10 using conditionally gene-deficient mice and queried their contribution to the zinc sparks of fertilization.

First, we confirmed the expression of zinc transporter in mouse oocytes. Our results are consistent with previous studies (*Kong et al., 2014*; *Chen et al., 2023*). ZIPs, including ZIP6 and ZIP10, are transmembrane proteins (*Eide, 2004*; *Kambe et al., 2004*; *Lu and Fu, 2007*; *Schmitt-Ulms et al., 2009*). In primordial follicles, the ooplasm staining of ZIP10 we anticipate corresponds to ooplasmic vesicular sites. ZIP10 expression shifted to the plasma membrane in primary and antral follicle phase. A similar localization shift of ZIP10 to the oocyte surface was reported during oocyte maturation, which is much later than reported here. Furthermore, in that study, ZIP10 was detected in the nuclear/nucleolar positions of oocytes of all follicular stages (*Chen et al., 2023*), which we did not observe. On the other hand, ZIP6 was expressed at the nuclear/nucleolar regions and in granulosa cells. This localization of ZIP6 was consistent with that in a previous study (*Chen et al., 2023*). However, our results failed to notice that ZIP6 shifts to the plasma membrane in antral follicle oocytes (*Kong et al.,*

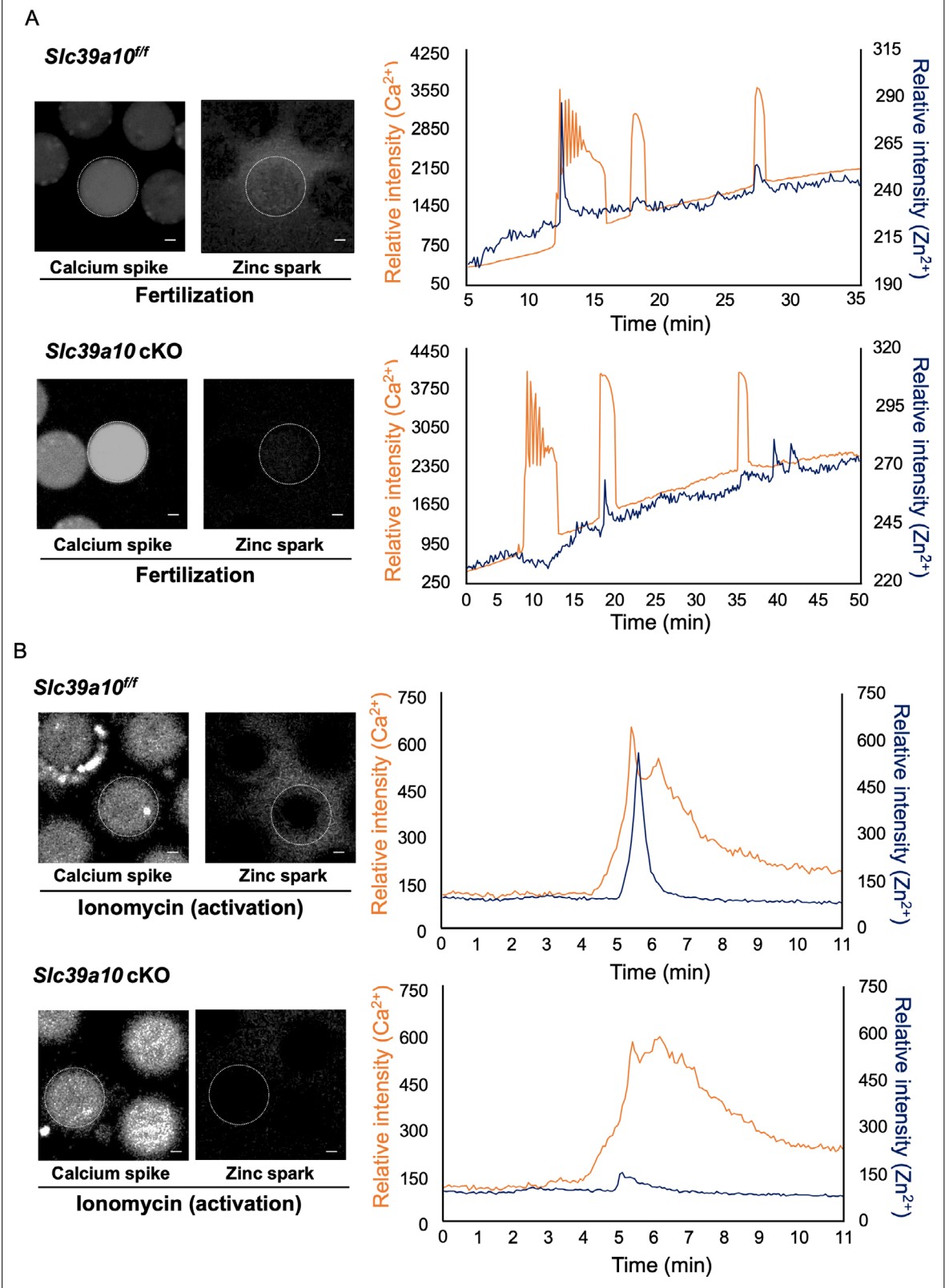

**Figure 3.** Measurement of calcium spike and zinc spark in *Slc39a10* cKO mice. (**A**) The representative images of calcium spike and zinc spark after in vitro fertilization (IVF) in mouse oocytes. Left side images showed calcium spikes. Right side images showed zinc sparks. The oocytes increased calcium ion and released zinc ion shortly after fertilization. The white dotted circles indicate the positions of oocytes. Successful fertilization was confirmed by simultaneously monitoring intracellular calcium oscillations with Calbryte 590 AM and extracellular zinc ions with FluoZin-3 every 4 s. Capacitated

*Figure 3 continued on next page*

*Figure 3 continued*

frozen-thawed sperm was added to metaphase II (MII) at 2 min after imaging start. Orange line showed calcium ion, and dark blue line showed zinc ion. Intracellular calcium increases immediately before a zinc spark. Scale bars denote 20 μm. (**B**) The representative images of an MII egg activated with 5 μM ionomycin followed by monitoring of intracellular calcium oscillations with Calbryte 590 AM and extracellular zinc using 20 μM FluoZin-3. The ionomycin was added to MII at 2 min after imaging start. Orange line showed calcium ion, and dark blue line showed zinc ion. Intracellular calcium increases immediately before a zinc spark. Scale bars denote 20 μm.

The online version of this article includes the following video and figure supplement(s) for figure 3:

**Figure supplement 1.** Measurement of calcium spike and zinc spark in *Slc39a6* cKO mice.

**Figure 3—video 1.** Monitoring of intracellular calcium ions during fertilization of *Slc39a6*$^{f/f}$ oocytes.
https://elifesciences.org/articles/106616/figures#fig3video1

**Figure 3—video 2.** Monitoring of extracellular zinc ions during fertilization of *Slc39a6*$^{f/f}$ oocytes.
https://elifesciences.org/articles/106616/figures#fig3video2

**Figure 3—video 3.** Monitoring of intracellular calcium ions during fertilization of *Slc39a6* cKO oocytes.
https://elifesciences.org/articles/106616/figures#fig3video3

**Figure 3—video 4.** Monitoring of extracellular zinc ions during fertilization of *Slc39a6* cKO oocytes.
https://elifesciences.org/articles/106616/figures#fig3video4

*2014*; *Chen et al., 2023*). The results indicate that ZIP6 may be fulfilling a distinct function within the oocytes compared to ZIP10.

To assess the role of these transporters in mice, we generated oocyte-specific *Slc39a6 or Slc39a10* knockout mice (*Figure 2—figure supplement 1*; *Slc39a6* cKO and *Slc39a10* cKO, respectively). Previous reports noted that the disruption of these transporters using specific morpholinos or incubation in function-blocking antibodies induced the change of intracellular labile zinc quota into mice oocytes (*Kong et al., 2014*). The amount of labile zinc ions in those mouse oocytes was measured using Fluozin-3AM, a zinc indicator. The amount of labile zinc ions in *Slc39a10* cKO oocytes was significantly lower than in the *Slc39a10*$^{f/f}$ (*Figure 2D*). We failed to observe any effect on zinc levels in *Slc39a6* cKO oocytes (*Figure 2—figure supplement 2B*). The results indicate that ZIP10 is mostly

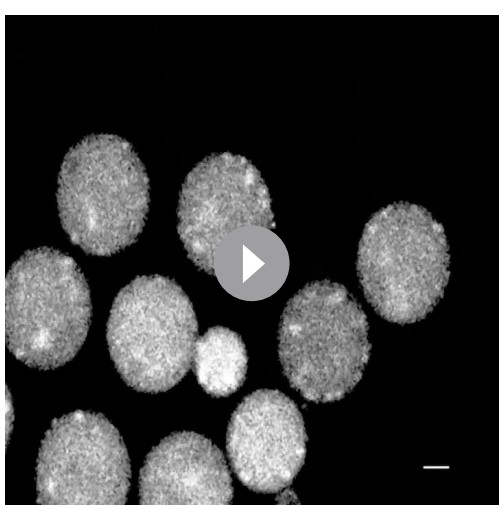

**Video 1.** Monitoring of intracellular calcium ions during fertilization of *Slc39a10*$^{f/f}$ oocytes. The changes in Ca$^{2+}$ were detected by Calbryte 590 AM every 4 s. *Slc39a10*$^{f/f}$ oocytes displayed the calcium oscillations following fertilization. The changes in zinc ions were also monitored simultaneously (*Video 2*). The video was excerpted from a maximum of 50 min. The scale bar represents 20 μm.
https://elifesciences.org/articles/106616/figures#video1

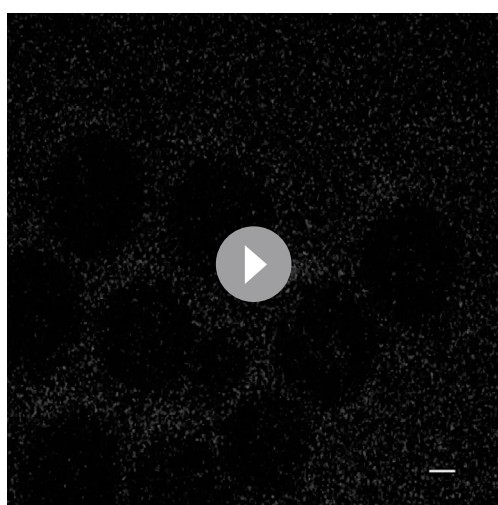

**Video 2.** Monitoring of extracellular zinc ions during fertilization of *Slc39a10*$^{f/f}$ oocytes. The changes in Zn$^{2+}$ were detected by FluoZin-3 every 4 s. *Slc39a10*$^{f/f}$ oocytes released zinc ions into the extracellular environment through a zinc spark following the first calcium rise following fertilization. The changes in calcium ions were also monitored simultaneously (*Video 1*). The video was excerpted from a maximum of 50 min. The scale bar represents 20 μm.
https://elifesciences.org/articles/106616/figures#video2

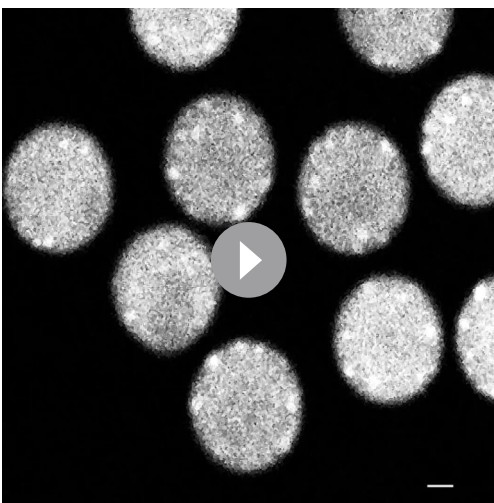

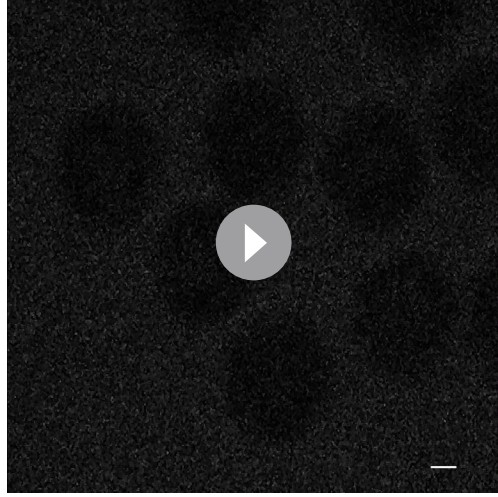

**Video 3.** Monitoring of intracellular calcium ions during fertilization of *Slc39a10* cKO oocytes. The changes in Ca²⁺ were detected by Calbryte 590 AM every 4 s. *Slc39a10* cKO oocytes displayed the calcium oscillations following fertilization. The changes in zinc ions were also monitored simultaneously (*Video 4*). The video was excerpted from a maximum of 50 min. The scale bar represents 20 μm.

https://elifesciences.org/articles/106616/figures#video3

**Video 4.** Monitoring of extracellular zinc ions during fertilization of *Slc39a10* cKO oocytes. The changes in Zn²⁺ were detected by FluoZin-3 every 4 s. *Slc39a10* cKO oocytes did not release zinc ions into the extracellular environment following the first calcium rise following fertilization. The changes in calcium ions were also monitored simultaneously (*Video 3*). The video was excerpted from a maximum of 50 min. The scale bar represents 20 μm.

https://elifesciences.org/articles/106616/figures#video4

responsible for the uptake of zinc ions in mouse oocytes.

Zinc-insufficient GV oocytes do not maintain the meiotic arrest at PI (*Kong et al., 2012*). This is suggested to happen by premature activation of the MOS-MAPK pathway in the presence of low zinc (*Kong et al., 2012*). In addition, low ooplasmic zinc accelerates meiotic progression that contributes to the extrusion of large polar bodies (*Bernhardt et al., 2011*). The early mitotic inhibitor 2 (EMI2), a zinc-binding APC/C proteasome inhibitor, is also an essential component of the cytoplasmic factor that initiates entry into MII phase. EMI2 is a zinc-binding protein, and when zinc is reduced, the activity of the APC/C proteasome is stimulated, increasing the degradation of CCNB1, reduced MPF activity leading to early release of meiosis arrest (*Bernhardt et al., 2011*; *Bernhardt et al., 2012*; *Suzuki et al., 2010*; *Shoji et al., 2014*; *Ohe et al., 2010*). Previous studies reported that targeted disruption of *Slc39a10* using morpholino injections and function-inhibiting antibodies during meiotic maturation perturbed meiosis progression and resulted in cell cycle arrest at the telophase I-like state (*Kong et al., 2014*). Based on these findings, we hypothesized that the reduced state of labile zinc ions in *Slc39a10* cKO mouse oocytes may resemble the state of zinc-deficient oocytes. We investigated the maturation progression and spindle organization in oocytes matured in vitro from the GV to MII stage. We failed to observe any differences (*Figure 2A, B, and C*). We also did not observe abnormalities in oocytes from *Slc39a6^f/f* mice (*Figure 2—figure supplement 2A*; p>0.05). We speculate that the disparate outcomes observed in previous studies may be attributed to the presence of trace amounts of labile zinc ions in media or in the oocytes.

Surprisingly, the amount of labile zinc ions in *Slc39a10* cKO 2PN zygote was higher than in *Slc39a10^f/f*, and despite some differences in fluorescence intensity between *Slc39a10* cKO GV, MII, and 2PN zygote, the labile zinc concentrations were not significantly different between these groups (*Figure 2D*). Therefore, we monitored zinc sparks and calcium oscillations in the *Slc39a10*-KO oocytes. Several studies reported the importance of IP₃Rs in mammalian oocytes (*Fissore et al., 1999*; *Parrington et al., 1998*), which is essential for egg activation because its inhibition precludes Ca²⁺ oscillations (*Miyazaki and Ito, 2006*; *Miyazaki et al., 1992*; *Xu et al., 2003*). It has been reported that a putative zinc-finger motif in a helical linker (LNK) domain near the C-terminus of IP₃R1 plays a role in IP₃R1 function (*Fan et al., 2015*; *Paknejad and Hite, 2018*). Recently, Akizawa et al. reported

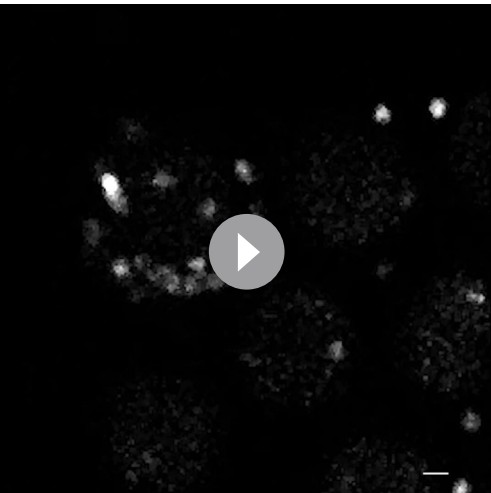

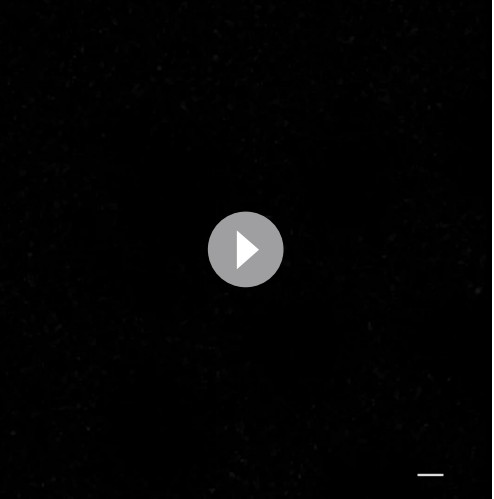

**Video 5.** Monitoring of intracellular calcium ions during parthenogenesis of *Slc39a10^{f/f}* oocytes. The changes in Ca²⁺ were detected by Calbryte 590 AM every 4 s. *Slc39a10^{f/f}* oocytes displayed a transient rise of calcium ions following artificial oocyte activation with ionomycin. The changes in zinc ions were also monitored simultaneously (**Video 6**). The video was excerpted from a maximum of 50 min. The scale bar represents 20 µm.

https://elifesciences.org/articles/106616/figures#video5

**Video 6.** Monitoring of extracellular zinc ions during parthenogenesis of *Slc39a10^{f/f}* oocytes. The changes in Zn²⁺ were detected by FluoZin-3 every 4 s. *Slc39a10^{f/f}* oocytes released zinc ions into the extracellular environment through a zinc spark following a transient calcium rise following artificial oocyte activation with ionomycin. The changes in calcium ions were also monitored simultaneously (**Video 5**). The video was excerpted from a maximum of 50 min. The scale bar represents 20 µm.

https://elifesciences.org/articles/106616/figures#video6

that both deficient and excessive zinc ions compromise IP₃R1 sensitivity, diminishing and terminating calcium oscillations (*Akizawa et al., 2023*). Our results showed that a single zinc spark occurs immediately after the first calcium rise of oscillations in *Slc39a10^{f/f}* oocytes, as reported by *Kim et al., 2011*; (*Figure 3A*), and similar results are obtained in *Slc39a6* cKO and *Slc39a6^{f/f}* mouse oocytes (*Figure 3—figure supplement 1*). However, in *Slc39a10* cKO oocytes, despite the presence of calcium elevations, zinc sparks failed to occur. Similarly, *Slc39a10* cKO oocytes activated with ionomycin (artificial oocyte activation) did not show a zinc spark (*Figure 3B*). In this study, calcium oscillations occurred in oocytes with low zinc state. In addition, there was no difference in the amplitude frequency of calcium ions in oocytes of both groups within the observation time. Akizawa et al. produced zinc deficiency state with TPEN, a strong chelator of intracellular labile zinc ions (*Akizawa et al., 2023*). On the other hand, we did not completely remove intracellular labile zinc ions from mouse oocytes. We speculate that zinc ions existed in *Slc39a10* cKO mouse oocytes, inducing Ca²⁺ release without compromising IP₃R1 sensitivity. However, this study demonstrated that the accumulation of intracellular zinc ions mediated by ZIP10 is essential for inducing zinc sparks in mouse oocytes.

The total cellular zinc ion content required for meiosis in the oocyte substantially increases (a 50% increase) from the prophase I arrest to the arrest at metaphase of meiosis II (*Kim et al., 2010*; *Bernhardt et al., 2011*). The zinc is stored in undefined cortical granules, and an average of 10⁶ zinc atoms is released from these vesicles at the time of fertilization (exocytosis) (*Kim et al., 2011*; *Que et al., 2015*; *Kong et al., 2014*; *Que et al., 2019*; *Jo et al., 2019*; *Lee et al., 2020*). There are several possibilities that might explain why *Slc39a10* conditional KO mice have low zinc sparks. The most logical explanation is that these oocytes experience reduced zinc loading of cortical zinc vesicles, leading to the absence of sparks. Our results suggest that in oocytes, zinc sparks and exocytosis require adequate concentrations of labile zinc ions in the cortical secretory vesicles, which is not accomplished when ZIP10 is not present.

Zinc sparks have been reported to contribute to the rapid and permanent block mechanisms to prevent polyspermy. The former is achieved by zinc in the extracellular milieu decelerating the forward motility of sperm (*Tokuhiro and Dean, 2018*). Second, the release of labile pools of zinc

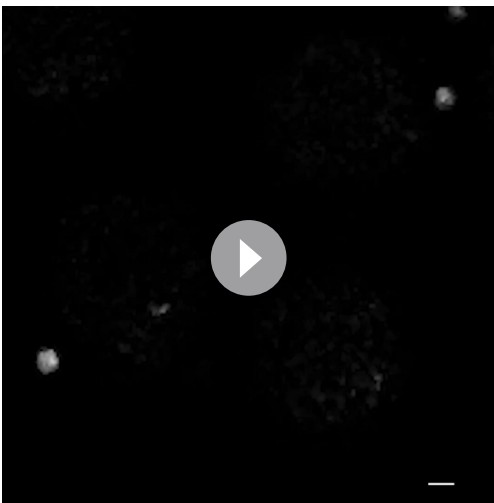

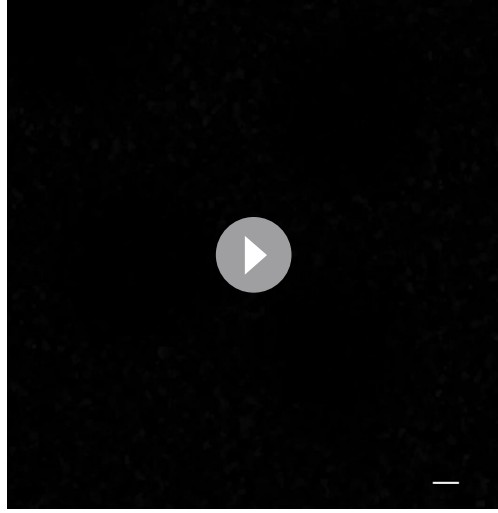

**Video 7.** Monitoring of intracellular calcium ions during parthenogenesis of *Slc39a10* cKO oocytes. The changes in Ca²⁺ were detected by Calbryte 590 AM every 4 s. *Slc39a10* cKO oocytes displayed a transient rise of calcium ions following artificial oocyte activation with ionomycin. The changes in zinc ions were also monitored simultaneously (*Video 8*). The video was excerpted from a maximum of 50 min. The scale bar represents 20 μm.

https://elifesciences.org/articles/106616/figures#video7

**Video 8.** Monitoring of extracellular zinc ions during parthenogenesis of *Slc39a10* cKO oocytes. The changes in Zn²⁺ were detected by FluoZin-3 every 4 s. *Slc39a10* cKO oocytes did not release zinc ions into the extracellular environment following a transient calcium rise following artificial oocyte activation with ionomycin. The changes in calcium ions were also monitored simultaneously (*Video 5*). The video was excerpted from a maximum of 50 min. The scale bar represents 20 μm.

https://elifesciences.org/articles/106616/figures#video8

at the same time that the zinc metalloendopeptidase ovastacin is released with the cortical granules (*Tokuhiro and Dean, 2018*; *Burkart et al., 2012*; *Schmitz et al., 2021*) causes ZP2 protein cleavage inducing a complete, although belated block to polyspermy (*Que et al., 2017*; *Aonuma et al., 1981*; *Tokuhiro and Dean, 2018*). However, the physiological significance of the zinc sparks' contributions to polyspermy has not been thoroughly tested, and *Slc39a10* cKO mice, which are almost devoid of them, offer a great model. Our results showing normal rates of monospermic fertilization in *Slc39a10* cKO *oocytes* suggest that contributions of zinc to polyspermy control are negligible (*Figure 4A*). Our results also show that the ovastacin undergoes normal release in *Slc39a10^f/f* oocytes after fertilization (*Figure 4E and D*), confirming previous studies of a minor role of the ZP2 to polyspermy. We observed the disappearance of JUNO after fertilization (*Bianchi et al., 2014*) *Slc39a10* cKO, suggesting that gamete fusion took place (*Figure 4F*). Together, these results support the view that zinc spark is not directly involved in the polyspermy rejection mechanism.

Interestingly, there were no differences in the rates of fertilization (pronucleus formation), but development to the blastocyst stage was significantly reduced in *Slc39a10*-null embryos (*Figure 4B*). Most of the embryos were arrested at the 2-cell stage and not developed beyond the 4-cell stage. Zinc insufficiency caused altered chromatin structure in the nuclei of blastomeres that displayed decreased global transcription, causing arrest in embryonic development (*Kong et al., 2015*). The abnormal zinc homeostasis, particularly during the 1-cell stage, inhibits the activation of the embryonic genome that occurs around the 2-cell stage in mice due to reduced translational capacity through the inhibition of ribosomal RNA synthesis by RNA polymerase I (*Chanfreau, 2013*; *Garner et al., 2021*). Our results also show that *Slc39a10* cKO blastocysts had significantly lower cell numbers compared to the blastocysts from *Slc39a10^f/f* mice (*Figure 4C*). Our results suggest that *Slc39a10* cKO mice oocytes can be fertilizable, but their developmental potential is decreased. The lack of ZIP10-mediated zinc influx during the folliculogenesis of *Slc39a10* cKO may compromise gene expression during these early stages of oocyte development. *Slc39a10* cKO mouse oocytes thus may have compromised developmental potential from the outset. Future studies should assess the transcriptomic or proteomic profile of *Slc39a10* cKO mouse oocytes.

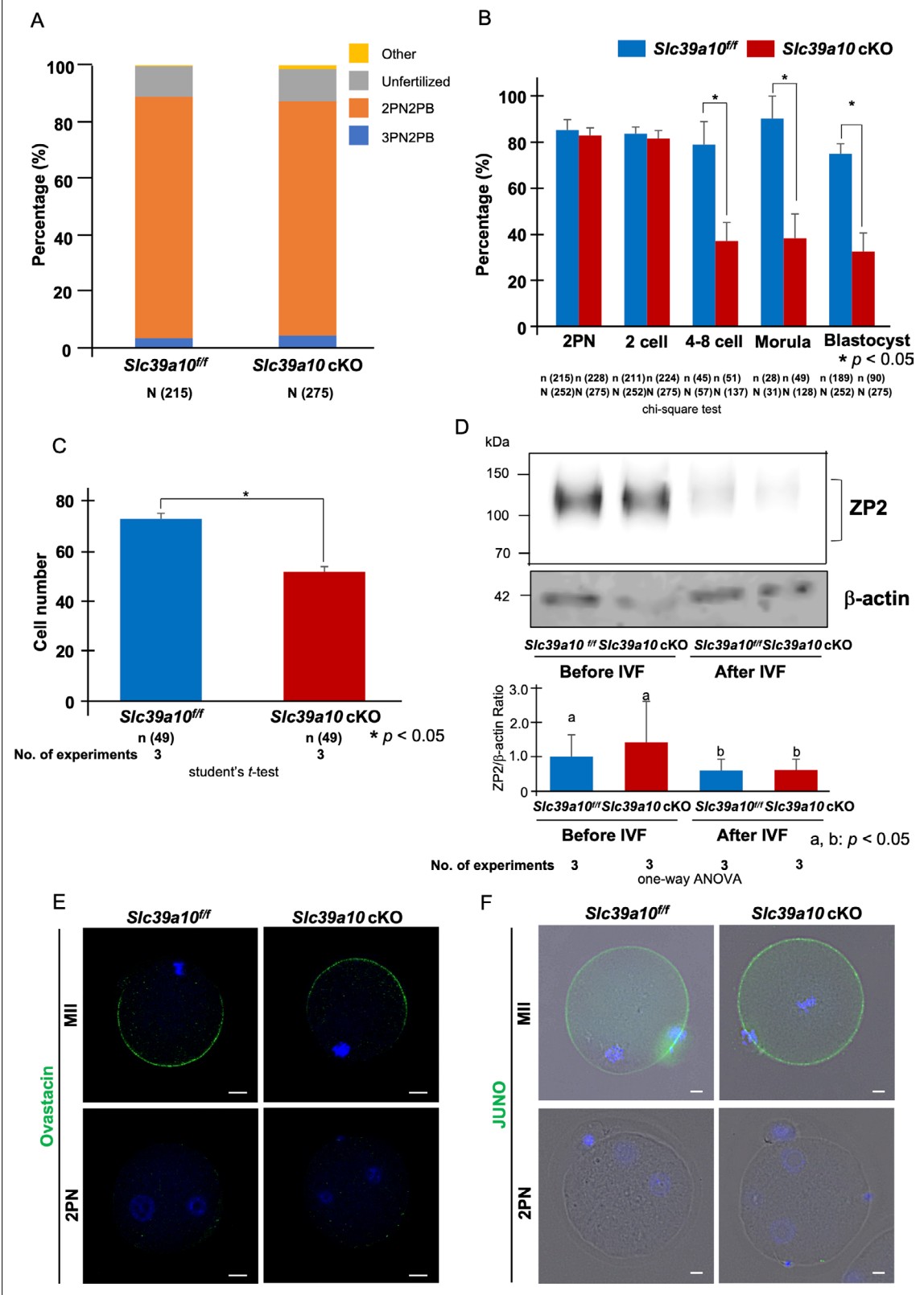

**Figure 4.** Presence of a mechanism to prevent multisperm fertilization in *Slc39a10* cKO mice. (**A**) The percentages of oocytes with each number of PN at 6 hr after insemination. Yellow region showed others, including degeneration, degression, and fragmentation. Gray region showed unfertilization, namely metaphase II (MII) oocytes. Orange showed 2PN2PB, namely embryo possessed one female and male pronucleous (2PN) and second polar body (2PB). Blue region showed multisperm fertilization (3PN2PB). (**B**) The percentage of fertilized oocytes and developmental embryos. Data represent the

*Figure 4 continued on next page*

*Figure 4 continued*

average ± SE of the experiments. The embryo development was observed at 6 (2PN), 24 (2-cell), 48 (4- to 8-cell), 72 (morula), and 96 (blastocyst) hours after IVF. The oocytes used for IVF were calculated as the parameter for the fertilization rate and the rate of embryo development. These experiments were repeated at least thrice. Statistical differences were calculated according to the chi-square test. Different letters represent significant differences (p<0.05). (C) The cell number of blastocyst derived from IVF. Blastocysts were fixed, immunostained, and physically flattened between a slide and coverslip. Photographs represent a single plane of focus. Nuclei are indicated by DAPI staining. The count used an inverted fluorescence microscope. These experiments were repeated three times, and each group counted a total of 46 embryos. Statistical differences were calculated according to the Student's *t*-test (p<0.05; significant difference). (D) Western blot of oocytes from *Slc39a10^{f/f}* and *Slc39a10* cKO mice at 0 or 6 hr after insemination using rat anti-ZP2 antibody. Intact ZP2 and the cleaved C-terminal fragment of ZP2 measured 120–130 kD and undetected, respectively. Expression level of β-actin serves as a protein loading control and quantified the expression level of ZP2. Molecular mass is indicated at the left. Statistical differences were calculated according to the one-way ANOVA. Different letters represent significant differences (p<0.05). (E) MII oocytes and 2PN embryos from *Slc39a10^{f/f}* and *Slc39a10* cKO mice were imaged by confocal microscopy after staining with rabbit anti-ovastacin (green). Chromosomes were stained with DAPI (blue). The scale bar represents 10 μm. (F) MII oocytes and 2PN embryos from *Slc39a10^{f/f}* and *Slc39a10* cKO mice were imaged by BZ-X700 microscopy after staining with rat anti-mouse FR4 (JUNO; green). Chromosomes were stained with DAPI (blue). The scale bar represents 10 μm.

The online version of this article includes the following source data and figure supplement(s) for figure 4:

**Source data 1.** PDF file containing original western blots for *Figure 4D*, indicating the relevant bands and treatments.

**Source data 2.** Original files for western blot analysis displayed in *Figure 4D*.

**Figure supplement 1.** Comparison of JUNO expression in *Slc39a10^{f/f}* and *Slc39a10* cKO mouse metaphase II (MII) oocytes.

**Figure supplement 2.** Percentage distribution of pronucleous in *Slc39a6* cKO mice and the number of fertilized oocytes and developmental embryos.

In conclusion, we elucidated that ZIP10 is required for zinc ion uptake into oocytes, and the intracellular zinc ions regulated by ZIP10 were important for zinc sparks and progression of embryonic development. However, the role of ZIP6 remained uncertain. Additionally, the absence of mechanistic insight for zinc spark and the inability to distinguish between the developmental and fertilization stage roles of ZIP10 remain unresolved. These challenges necessitate further investigation. Currently, many infertility patients exist in the world (the survey of WHO). It is reported that approximately half of the adult females are deficient in zinc in their serum (*Prasad, 1996*; *Yokokawa et al., 2020*). Zinc has been reported to play essential roles in many physiological functions, including reproductive functions (*Suzuki et al., 2021*, *MacDonald, 2000*; *Jackson et al., 2008*; *Bernhardt et al., 2012*, *Suzuki et al., 2010*, *Ohe et al., 2010*). The findings will help elucidate the role of zinc homeostasis in the fields of fertilization/embryogenesis. Furthermore, the development of zinc-focused assisted reproductive technologies and fertilization/embryogenesis media could contribute to improving the developmental potential of oocytes and embryos in other mammalian species.

## Materials and methods

All chemicals and reagents were purchased from Sigma-Aldrich (St. Louis, MO, USA) unless otherwise stated.

### Animals

Animals were housed in the barrier facility at Azabu University. The deletion of the ZIP10 gene in the whole body is known to result in embryonic lethality (*He et al., 2023*). The conditional knockout mice were generated by the Cre-loxP system. The following mouse strains were used: *Slc39a10^{flox/flox}* (*Slc39a10^{f/f}*) mouse <B6;129-Slc39a10<tm1.1Tfk>> (RBRC06221) (*Hojyo et al., 2014*; *Miyai et al., 2014*; *Bin et al., 2017*) was provided by the RIKEN BRC through the National BioResource Project of the MEXT/AMED, Japan. *Slc39a6^{flox/flox}* (*Slc39a6^{f/f}*) mouse <B6;129-Slc39a6> was generated by AdAMS. *Gdf9^{Cre/+}* mice <Stock Tg (GDF9-iCre) 5092Coo/J, Stock No: 011062> were delivered from Jackson Laboratory (*Lan et al., 2004*). Crlj: C57BL/6J female mice (6–8 weeks of age) and Crlj: C57BL/6J male mice (8–12 weeks of age) were purchased from Charles River Laboratories Japan (Yokohama, Japan). The mice were housed under controlled lighting conditions (daily light period, 0600–1800 hr). The study was approved by the Animal Experimentation Committees of Azabu University and was performed in accordance with the committees' guidelines (200318-13 and 230309-1). Oocyte-specific gene knockout (*Slc39a10^{f/f}*, *Gdf9^{Cre/+}*; *Slc39a10* cKO) mice were generated by crossing *Gdf9^{Cre/+}* male mice with *Slc39a10^{f/f}* female mice (*Figure 2—figure supplement 1*). Similarly, *Slc39a6* cKO mice were generated (*Figure 2—figure supplement 1*).

## In situ hybridization

ISH was performed as previously described with some modifications (*Namiki et al., 2023*). Briefly, fixed ovaries were paraffin-embedded and paraffin sections (6 µm) were mounted on MAS-coated slides (Matsunami Glass Industries, Osaka, Japan) under RNase-free conditions. Sense or antisense digoxigenin (DIG)-labeled RNA probes for *Slc39a10/Zip10* were purchased from Genostaff. The sections were deparaffinized, rehydrated, and postfixed in 10% neutral buffered formalin for 30 min at 37°C, followed by the treatment with 0.2% hydrogen chloride and 5 µg/ml proteinase K (FUJIFILM Wako Pure Chemical, Osaka, Japan) for 10 min at 37°C, respectively. Hybridization was performed with DIG-labeled probes (250 ng/ml) in a humidified chamber at 60°C overnight. The slides were washed after hybridization, then treated with blocking reagent (Genostaff) for 15 min and alkaline phosphatase-conjugated anti-DIG antibody (1:2,000; Roche Diagnostics, Basel, Switzerland) for 1 hr at room temperature. The signals were detected by 4-nitro-blue tetrazolium/5-bromo-4-chloro-3-indolyl phosphate (NBT/BCIP, Roche Diagnostics) in a humidified container for 12 hr at 4°C. The sections were counterstained with Kernechtrot solution (Muto Pure Chemicals, Tokyo, Japan). Signals detected by the sense probe were used as a control for background levels.

## IF staining for ovary

IF was performed as previously described with some modifications (*Namiki et al., 2023*). Fixed ovaries were paraffin-embedded and the sections (6 µm) were deparaffinized, hydrated, and conducted for antigen retrieval by autoclaving in 10 mM sodium citrate buffer (pH = 6.0) for 5 min. The sections were further incubated on ice for 30 min. After blocking with Bloking One Histo (06349-64, NACALAI TESQUE Inc, Kyoto, Japan) for 1 hr, the slides were incubated with primary antibody for Rabbit anti-mouse-ZIP10 (1:200; *Miyai et al., 2014*), Rabbit anti-SLC39A6 (1:200, HPA042377, Sigma), Rabbit anti-Foxl2 (1:300; *Cocquet et al., 2002*; *Polanco et al., 2010*), Rat anti-ZP2 (1:100, sc-32752, Santa Cruz Biotechnology, Dallas, TX, USA) in Can Get Signal immunostain (TOYOBO, Tokyo, Japan) overnight at 4°C. The slides were incubated with Alexa Fluor 488 donkey anti-rabbit IgG (H+L), Alexa Fluor 594 donkey anti-rabbit IgG (H+L), or Alexa Fluor 488 donkey anti-rat IgG (H+L) conjugated secondary antibodies (Jackson Immuno Research Laboratories, West Grove, PA, USA) diluted 1:500 in Can Get Signal immunostain (TOYOBO) for 1 hr, and mounted with ProLong Glass Antifade Mountant with NucBlue Stain (P36981, Thermo Fisher Scientific, Waltham, MA, USA). Micrographs were captured by BZ-X700 microscopy (Keyence, Osaka, Japan).

## Oocyte preparation

GV oocytes were collected in the manner described in our previous study (*Ito et al., 2008*) with some modifications. Ovaries were collected from female mice 48 hr after intraperitoneal injection with 5 IU equine chorionic gonadotropin (eCG) (PMS; Nippon Zenyaku Kogyo, Fukushima, Japan). They were placed in a 35 mm culture dish containing MEMα (no nucleosides, powder; Gibco/Thermo Fisher Scientific, Tokyo, Japan) medium with 26 mM NaHCO$_3$, 75 mg/l penicillin, 50 mg/l streptomycin sulfate, 5% (vol/vol) heat-treated fetal calf serum (FCS), and 10 ng/ml epidermal growth factor (EGF). Cumulus-oocyte complexes (COCs) were released from the antral follicles by gentle puncturing with a needle.

To obtain MII oocytes, the mice were intraperitoneally injected with 5 IU eCG followed by injection with 5 IU human chorionic gonadotropin (hCG) (Gonatropin; ASKA Pharmaceutical, Tokyo, Japan) at 48 hr later. COCs were collected from the oviductal ampulla 14–16 hr after hCG injection.

These oocytes were used in the following experiments.

## Count of ovulated oocytes

COCs-MII were obtained by superovulation treatment. The cumulus cells were removed from the COCs-MII with hyaluronidase (1 mg/ml) and gentle pipetting. All collected oocytes were counted and calculated as the number of ovulations.

## IVM of GV oocytes

IVM was conducted in the manner described in our previous study (*Kamoshita et al., 2021*) with some modifications. Ovaries were placed in a 35 mm culture dish containing MEMα medium with 26 mM NaHCO$_3$, 75 mg/l penicillin, 50 mg/l streptomycin sulfate, 5% (vol/vol) FCS, and 10 ng/ml EGF.

COCs-GV were released from the antral follicles by gentle puncturing with a needle. The COCs-GV were washed three times and cultured in 500 µl of the same medium in a four-well dish at 37°C in an atmosphere of 5% $CO_2$ in air for 10, 12, and 14 hr. After the culture, cumulus cells were removed from the COCs with hyaluronidase (1 mg/ml) and gentle pipetting. The extrusion of the first polar body was evaluated at each time.

## IF for oocytes/preimplantation embryos

The α-tubulin and JUNO (PE anti-mouse FR4) were performed with some modification of the methods of *Inoue et al., 2017*. The oocytes or embryos were fixed in 4% PFA for 30 min at room temperature, washed in PBS containing 1% polyvinyl alcohol (PBS/PVA). They were permeabilized by treatment of 0.5% Triton X-100 for 15 min and washed two times in 1% BSA/PBS/PVA followed by blocking for 20 min in the same medium. They were incubated overnight at 4°C with primary antibodies to rabbit anti-α-tubulin (1:200; 11H10, Cell Signaling Technology) or rat anti-mouse FR4 (JUNO; 1:250; 12A5 BioLegend), washed three times in 1% BSA/PBS/PVA. Primary antibodies were detected using Alexa Fluor 488 donkey anti-rabbit IgG (H+L) (1:250) or Alexa Fluor 488 donkey anti-rat IgG (H+L) (1:250) for 1 hr at room temperature. After staining, all samples were mounted in VECTASHIELD Mounting Medium with DAPI (H-1200; Vector Laboratories, CA, USA) and imaged using BZ-X700 microscopy (Keyence, Osaka, Japan).

The ovastacin staining was performed with some modifications of the methods of *Burkart et al., 2012*. Oocytes or embryos were fixed in 4% PFA overnight at 4°C, washed in PBS containing 0.3% polyvinylpyrrolidone (PVP), and then blocked in 0.3% BSA/0.1 M glycine (three times for 10 min) followed by permeabilization in 0.2% Triton X-100 for 15 min (*Baibakov et al., 2007*). Samples were then incubated overnight at 4°C with rabbit polyclonal anti-ovastacin (*Burkart et al., 2012*; 1:200; gifted antibody), washed with 0.3% PVP/0.1% Tween (three times for 10 min), and incubated for 1 hr at room temperature with Alexa Fluor 488 anti-rabbit secondary antibody (1:500) followed by staining and mounting with VECTASHIELD Mounting Medium with DAPI. Samples were imaged using TCS SP5 II confocal microscope (Leica Microsystems, Wetzlar, Germany).

## In vitro fertilization

IVF and sperm collection were conducted with some modification of the method described in our previous study (*Kageyama et al., 2023*). In brief, ovulated COCs-MII were preincubated for 1 hr in 80 µl human tubal fluid (HTF) droplets supplemented with 1.25 mM reduced glutathione (GSH). Frozen-thawed sperm suspensions were suspended in 200 µl preincubation medium (HTF containing 0.4 mM methyl-β-cyclodextrin) and 0.1 mg/ml PVA, but without bovine serum albumin, and were incubated at 37°C under 5% $CO_2$ in humidified air for 1 hr. At the time of insemination, preincubated spermatozoa were transferred into the droplets with oocytes at final concentrations of $2.0 \times 10^6$ sperm/ml. After 6 hr, oocytes were separated from spermatozoa and cumulus cells using a fine glass pipette and transferred into 50 µl KSOMaa medium. They were cultured at 37°C under 5% $CO_2$ in humidified air for approximately 24–96 hr. The embryos were observed at 24, 48, 72, and 96 hr after IVF, and the number of 2-cell, 4- to 8-cell, morula, and blastocyst stage embryos was counted, respectively. The oocytes used for IVF were calculated as the parameter for the fertilization rate and the rate of embryo development. After IVF at 96 hr, the blastocysts were fixed in 4% PFA for 30 min at room temperature, followed by three times washes in PBS/PVA for 30 min each. Nuclear DNA was stained and mounted in VECTASHIELD Mounting Medium with DAPI. Cell numbers were determined by visually inspecting nuclei stained with DAPI using an inverted fluorescence microscope.

## Zinc measurements

FluoZin-3AM staining was conducted using the method described in our previous study (*Kageyama et al., 2022a*). The obtained GV oocytes and cumulus cells were removed from the COCs with gentle pipetting. The obtained MII oocytes and cumulus cells were removed from the COCs with hyaluronidase (1 mg/ml) and gentle pipetting. Oocytes with a polar body were defined as MII. After IVF for 6 hr, two pronucleus stage embryos (2PN) were collected. GV, MII, and 2PN were loaded in 50 µl medium that was suitable for each stage containing the amyl ester of the membrane permeant zinc-specific fluorophore, FluoZin-3AM (2 µM; F24195, Thermo Fisher Scientific, excitation 494 nm/emission 516 nm) for 1 hr in humidified $CO_2$ (5% [vol/vol] in air) at 37°C followed by washing three times

in medium and then observation with TCS SP5 II confocal microscope. FluoZin-3 has been extensively characterized for measurements of free intracellular zinc in live cells using microscopy and has an affinity constant (Kd) for zinc of 15 nM (*Gee et al., 2002a*; *Gee et al., 2002b*). Our previous study showed MII oocytes were treated with FluoZin-3 AM for 60 min, the change of fluorescence was confirmed in the cytoplasm of the oocytes and embryos, suggesting this treatment duration with FluoZin-3 AM is suitable for detection of zinc ions in oocytes and embryos (*Kageyama et al., 2022b*). The pixel intensity per unit area after background subtraction was determined in GV, MII, and 2PN within the circle (white circle) and ImageJ image-processing software.

## Measurement of Ca²⁺ and zinc spark during fertilization

Fresh or frozen-thawed sperm suspension was suspended in 200 µl preincubation medium. The 1 µl (fresh) or 20 µl (frozen-thawed) sperm suspension was placed in 40–80 µl HTF drop in pre-insemination dish about 20 min before insemination. The zona pellucida was punctured by piezoelectric pulses applied to four locations with a 15 µm injection needle. The treated MII oocytes were cultured in CZB containing Calbryte 590 AM (10 µM, #20700, AAT Bioquest, CA, USA, excitation 581 nm/emission 593 nm) for 10 min in humidified $CO_2$ (5% [vol/vol] in air) at 37°C. While the oocytes are incubated for 10 min, make a drop of 10 µl of PVA(-), Ca(-) HEPES-CZB (H-CZB), and pull out 7 µl of medium from drop with pipette man in the insemination dish. After 10 min, the oocytes are washed and transferred to the insemination dish, and the oocytes are attached to the dish. After attachment, an HTF containing 7 µl of BSA and a membrane-impermeable zinc-specific fluorophore, FluoZin-3 (20 µM, F24194, Thermo Fisher Scientific, excitation 494 nm/emission 516 nm), is added to the 3 µl drop containing the oocytes gently using a capillary. About 1 µl sperm was sucked from the pre-insemination drop with the capillary under a stereomicroscope and placed into the drop containing the oocytes, to start imaging. In the case of ionomycin treatment, after calcium labeling, MII were allowed to settle in 45 µl PVA(-), Ca(-) H-CZB containing FluoZin-3 medium drop on dish, to start imaging. The 5 µl ionomycin (5 µM; #407950) was added to MII at 2 min after the start of imaging. Imaging was performed on a confocal microscope using 488 nm (Zn²⁺) and 555 nm (Ca²⁺) excitation (Nikon Solutions, Tokyo, Japan) for max 50 min every 4 s. Imaging analysis was performed by defining regions of interest (ROIs) and measuring fluorescence intensity over time using NIS-Elements (Nikon). The intracellular ROIs were drawn as the entire interior area of the cell. The extracellular ROIs were defined as a ring around the perimeter of the cell. The ring thickness was conserved for all data analyses.

## Western blotting

Western blotting was carried out as described (*Ito et al., 2010*) with some modifications. Thirty MII oocytes or 2PN embryos were lysed in Laemmli sample buffer (Bio-Rad Laboratories, Tokyo, Japan) with 5% 2-mercaptoethanol. Samples were separated on 8% Bis-Tris gels by SDS-PAGE and transferred to PVDF membranes (Bio-Rad). The PVDF membranes were blocked in 10% skim milk (FUJIFILM Wako) in Tris-buffered saline with 0.1% Tween-20 (Yoneyama Yakuhin Kogyo, Osaka, Japan) and probed with primary antibody to Rabbit anti-mouse-ZIP10 (1:1000; *Miyai et al., 2014*), Rabbit anti-SLC39A6 (1:1000, HPA042377), rat anti-ZP2 (1:1,000) or monoclonal mouse anti-β-actin (1:5000; A5316, Sigma-Aldrich) for overnight at 4°C. The membranes were incubated with a secondary antibody: HRP-conjugated anti-rabbit IgG (1:5000; Cell Signaling Technology, Danvers, MA, USA), HRP-conjugated anti-rat IgG (1:5000; Cell Signaling Technology), or HRP-conjugated anti-mouse IgG (1:5000; Cell Signaling Technology) for 1 hr at room temperature. After washing of the membranes, immunoreactive proteins were visualized using ECL Western Blotting Analysis System (Cytiva Global Life Sciences Technologies, Tokyo, Japan), according to the manufacturer's recommendation. The membranes were exposed by ImageQuant LAS 4000. After exposure, the membranes were incubated for 30 min at 50°C in the stripping buffer, including 1.5 M Tris-HCl (pH 6.8), 10% SDS, 2-mercaptoethanol followed by extensive washing of the membranes. The membranes were probed with primary antibody to mouse anti-β-actin (1:5000; Sigma-Aldrich) for overnight at 4°C. The membranes were incubated with secondary antibody; FRP-conjugated anti-mouse IgG (1:5000; Cell Signaling) for 1 hr at room temperature. The same procedure was followed below. The intensities of ZP2 bands were measured by quantitative analysis by densitometry using ImageJ.

## Statistical analysis

Values from three or more times were used for evaluation of statistical significance. Statistical analysis was performed using Statcel 3 software (OMS Ltd., Saitama, Japan). The fertility, the total numbers of oocytes collected from each mouse, the percentage of first polar extrusions, the fluorescence intensity of Fluozin-3AM, the cell count of blastocysts, and quantification of JUNO expression were evaluated statistically by Student's or Welch's *t*-test analysis. The rate of fertilization and embryo development was analyzed using chi-square tests. The quantification of ZP2 expression before and after fertilization was analyzed using one-way ANOVA. Values are shown as means ± SEM, and significant differences were considered at p-values<0.05.

## Acknowledgements

We thank the members of the Laboratory of Animal Reproduction, School of Veterinary Medicine, Azabu University, for technical help. *Slc39a10*flox/flox (*Slc39a10*f/f) mouse <B6;129-Slc39a10<tm1.1Tfk>> (RBRC06221) was provided by the RIKEN BRC through the National BioResource Project of the MEXT/AMED, Japan. We also thank Dr. Austin J Cooney (The University of Texas at Austin, Austin, United States) for providing GDF9-iCre mice. This research was supported by Grants-in-Aid for Scientific Research from the Japan Society for the Promotion of Science (JSPS) (KAKENHI, 21H02384 and 20H05373 to JI, 22H04268, 24H02601, and 24K23214 to AK), JSPS KAKENHI Grant Number JP 15H04584 (AdAMS) to JI, the Sasakawa Scientific Research Grant from The Japan Science Society (2019-4044) to AK and Mishima Kaiun Memorial Foundation (A-135) to AK. This work was supported by JSPS KAKENHI Grant Number JP 22H04922 (AdAMS). This study was also supported by the Center for Diversity, Equity & Inclusion, Azabu University (AK). This research was partially supported by the Center for Human and Animal Symbiosis Science, Azabu University, and a research project grant awarded by the Azabu University Research Services Division to JI.

## Additional information

### Funding

| Funder | Grant reference number | Author |
|---|---|---|
| Japan Society for the Promotion of Science | 21H02384 | Junya Ito |
| Japan Society for the Promotion of Science | 20H05373 | Junya Ito |
| Japan Society for the Promotion of Science | 22H04268 | Atsuko Kageyama |
| Japan Society for the Promotion of Science | 24H02601 | Atsuko Kageyama |
| Japan Society for the Promotion of Science | 24K23214 | Atsuko Kageyama |
| Japan Society for the Promotion of Science | 15H04584 | Junya Ito |
| Japan Science Society | 2019-4044 | Atsuko Kageyama |
| Mishima Kaiun Memorial Foundation | A-135 | Atsuko Kageyama |
| Japan Society for the Promotion of Science | 22H04922 | Junya Ito |
| Azabu University | | Atsuko Kageyama Junya Ito |

The funders had no role in study design, data collection and interpretation, or the decision to submit the work for publication.

## Author contributions
Atsuko Kageyama, Conceptualization, Data curation, Investigation, Visualization, Writing – original draft, Writing – review and editing; Narumi Ogonuki, Investigation, Visualization, Methodology, Writing – review and editing; Takuya Wakai, Methodology, Writing – review and editing; Takafumi Namiki, Investigation, Visualization, Writing – review and editing; Yui Kawata, Investigation, Writing – review and editing; Manabu Ozawa, Yasuhiro Yamada, Resources, Writing – review and editing; Toshiyuki Fukada, Resources, Supervision, Writing – review and editing; Atsuo Ogura, Supervision, Methodology, Writing – review and editing; Rafael Fissore, Naomi Kashiwazaki, Supervision, Writing – review and editing; Junya Ito, Conceptualization, Supervision, Writing – review and editing

## Author ORCIDs
Atsuko Kageyama (ID) https://orcid.org/0009-0006-7486-5711
Takuya Wakai (ID) https://orcid.org/0000-0003-4705-8974
Atsuo Ogura (ID) https://orcid.org/0000-0003-0447-1988
Junya Ito (ID) https://orcid.org/0000-0001-9398-7358

## Ethics
The study was approved by the Animal Experimentation Committees of Azabu University and were performed in strict accordance with the committees' guidelines (200318-13 and 230309-1). Every effort was made to minimize suffering during all treatments for mice.

Reviewer #1 (Public review): https://doi.org/10.7554/eLife.106616.4.sa1
Author response https://doi.org/10.7554/eLife.106616.4.sa2

# Additional files

## Supplementary files
MDAR checklist

## Data availability
All data needed to evaluate the conclusions in the paper are present in the paper and/or the supplementary materials.

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
