## [Editor Report · eLife Assessment]

This study presents significant and novel insights into the roles of zinc in mammalian meiosis/fertilization events. These findings are **useful** to our understanding of these processes. The evidence presented is **solid**, with experiments being well-designed, carefully described, and interpreted with appropriate rigor. The authors acknowledge the lack of mechanistic insight which represents the main limitation of the study.

---

## [Referee Report · Reviewer #1 (Public review)]

The revised manuscript addresses several reviewer concerns, and the study continues to provide useful insights into how ZIP10 regulates zinc homeostasis and zinc sparks during fertilization in mice. The authors have improved the clarity of the figures, shifted emphasis in the abstract more clearly to ZIP10, and added brief discussion of ZIP6/ZIP10 interactions and ZIP10's role in zinc spark-calcium oscillation decoupling. However, some critical issues remain only partially addressed.

(1) Oocyte health confound: The use of Gdf9-Cre deletes ZIP10 during oocyte growth, meaning observed defects could result from earlier disruptions in zinc signaling rather than solely from the absence of zinc sparks at fertilization. The authors acknowledge this and propose transcriptome profiling as a future direction. However, since mRNA levels often do not accurately reflect protein levels and activity in oocytes, transcriptomics may not be particularly informative in this context. Proteomic approaches that directly assess the molecular effects of ZIP10 loss seem more promising. Although current sensitivity limitations make proteomics from small oocyte samples challenging, ongoing improvements in this area may soon allow for more detailed mechanistic insights.

(2) ZIP6 context and focus: The authors clarified the abstract to emphasize ZIP10, enhancing narrative clarity. This revision is appropriate and appreciated.

(3) Follicular development effects: The biological consequences of ZIP6 and ZIP10 knockout during folliculogenesis are still unknown. The authors now say these effects will be studied in the future, but this still leaves a major mechanistic gap unaddressed in the current version.

(4) Zinc spark imaging and probe limitations: The addition of calcium imaging enhances the clarity of Figure 3. However, zinc fluorescence remains inadequate, and the authors depend solely on FluoZin-3AM, a dye known for artifacts and limited ability to detect subcellular labile zinc. The suggestion that C57BL/6J mice may differ from CD1 in vesicle appearance is plausible but does not fully address concerns about probe specificity and resolution. As the authors acknowledge, future studies with more selective probes would increase confidence in both the spatial and quantitative analysis of zinc dynamics.

(5) Mechanistic insight remains limited: The revised discussion now recognizes the lack of detailed mechanistic understanding but does not significantly expand on potential signaling pathways or downstream targets of ZIP10. The descriptive data are useful, but the inability to pinpoint how ZIP10 mediates zinc spark regulation remains a key limitation. Again, proteomic profiling would probably be more informative than transcriptomic analysis for identifying ZIP10-dependent pathways once technical barriers to low-input proteomics are overcome.

Overall, the authors have reasonably revised and clarified key points raised by reviewers, and the manuscript now reads more clearly. However, the main limitation, lack of mechanistic insight and the inability to distinguish between developmental and fertilization-stage roles of ZIP10, remains unresolved. These should be explicitly acknowledged when framing the conclusions.

Comments on revisions: I have no further comments to add to this review.

---

## [Author Response]

The following is the authors’ response to the previous reviews

**Reviewer #1 (Public review):**
The revised manuscript addresses several reviewer concerns, and the study continues to provide useful insights into how ZIP10 regulates zinc homeostasis and zinc sparks during fertilization in mice. The authors have improved the clarity of the figures, shifted emphasis in the abstract more clearly to ZIP10, and added brief discussion of ZIP6/ZIP10 interactions and ZIP10's role in zinc spark-calcium oscillation decoupling. However, some critical issues remain only partially addressed.

Thank you for your valuable inputs. We plan to address the issues that could not be clarified in this report going forward.

(1) Oocyte health confound: The use of Gdf9-Cre deletes ZIP10 during oocyte growth, meaning observed defects could result from earlier disruptions in zinc signaling rather than solely from the absence of zinc sparks at fertilization. The authors acknowledge this and propose transcriptome profiling as a future direction. However, since mRNA levels often do not accurately reflect protein levels and activity in oocytes, transcriptomics may not be particularly informative in this context. Proteomic approaches that directly assess the molecular effects of ZIP10 loss seem more promising. Although current sensitivity limitations make proteomics from small oocyte samples challenging, ongoing improvements in this area may soon allow for more detailed mechanistic insights.

Thank you for your suggestions. We will keep that in mind for the future.

(2) ZIP6 context and focus: The authors clarified the abstract to emphasize ZIP10, enhancing narrative clarity. This revision is appropriate and appreciated.

Thanks to your feedback, my paper has improved. Thank you for your evaluation.

(3) Follicular development effects: The biological consequences of ZIP6 and ZIP10 knockout during folliculogenesis are still unknown. The authors now say these effects will be studied in the future, but this still leaves a major mechanistic gap unaddressed in the current version.

As you mentioned, we have not been able to clarify the effects of ZIP6 and ZIP10 knockout on follicle formation. The effects of ZIP6 and ZIP10 knockout on follicle formation will be discussed in the future.

(4) Zinc spark imaging and probe limitations: The addition of calcium imaging enhances the clarity of Figure 3. However, zinc fluorescence remains inadequate, and the authors depend solely on FluoZin-3AM, a dye known for artifacts and limited ability to detect subcellular labile zinc. The suggestion that C57BL/6J mice may differ from CD1 in vesicle appearance is plausible but does not fully address concerns about probe specificity and resolution. As the authors acknowledge, future studies with more selective probes would increase confidence in both the spatial and quantitative analysis of zinc dynamics.

Thank you for your comment. Moving forward, we plan to conduct spatial and quantitative analyses of zinc dynamics using various other zinc probes.

(5) Mechanistic insight remains limited: The revised discussion now recognizes the lack of detailed mechanistic understanding but does not significantly expand on potential signaling pathways or downstream targets of ZIP10. The descriptive data are useful, but the inability to pinpoint how ZIP10 mediates zinc spark regulation remains a key limitation. Again, proteomic profiling would probably be more informative than transcriptomic analysis for identifying ZIP10-dependent pathways once technical barriers to low-input proteomics are overcome.

Thank you for your helpful advice. I'll use it as a reference for future analysis.

Future studies should assess the transcriptomic or proteomic profile of Zip10^d/d^ mouse oocytes (P.11 Line 349-350).

Overall, the authors have reasonably revised and clarified key points raised by reviewers, and the manuscript now reads more clearly. However, the main limitation, lack of mechanistic insight and the inability to distinguish between developmental and fertilization-stage roles of ZIP10, remains unresolved. These should be explicitly acknowledged when framing the conclusions.

We have added the two limitations you pointed out to the conclusion section of the main text.

However, the role of ZIP6 remained uncertain. Additionally, the absence of mechanistic insight for zinc spark and the inability to distinguish between the developmental and fertilization stage roles of ZIP10 remain unresolved. These challenges necessitate further investigation (P.11-12 Line 354-357).